# Telemedicine in Cardiology: Modern Technologies to Improve Cardiovascular Patients’ Outcomes—A Narrative Review

**DOI:** 10.3390/medicina58020210

**Published:** 2022-02-01

**Authors:** Kamil Kędzierski, Jadwiga Radziejewska, Agnieszka Sławuta, Magdalena Wawrzyńska, Jacek Arkowski

**Affiliations:** 1Department of Medical Emergencies, Wrocław Medical University, ul. K. Parkowa 34, 51-616 Wrocław, Poland; kamil.kedzierski@umw.edu.pl; 2Klodzko County Hospital, ul Szpitalna 1a, 57-300 Kłodzko, Poland; sekretariat@zoz.klodzko.pl; 3Department of Internal and Occupational Diseases, Hypertension and Clinical Oncology, Wrocław Medical University, ul Borowska 213, 50-556 Wrocław, Poland; aslawuta@tlen.pl; 4Center of Preclinical Studies, Wrocław Medical University, ul. K. Bartla 5, 51-618 Wrocław, Poland; magdalena.wawrzynska@umw.edu.pl

**Keywords:** cardiology, telemedicine, remote, monitoring

## Abstract

The registration of physical signals has long been an important part of cardiological diagnostics. Current technology makes it possible to send large amounts of data to remote locations. Solutions that enable diagnosis and treatment without direct contact with patients are of enormous value, especially during the COVID-19 outbreak, as the elderly require special protection. The most important examples of telemonitoring in cardiology include the use of implanted devices such as pacemakers and defibrillators, as well as wearable sensors and data processing units. The arrythmia detection and monitoring patients with heart failure are the best studied in the clinical setting, although in many instances we still lack clear evidence of benefits of remote approaches vs. standard care. Monitoring for ischemia is less well studied. It is clear however that the economic and organizational gains of telemonitoring for healthcare systems are substantial. Both patients and healthcare professionals have expressed an enormous demand for the further development of such technologies. In addition to these subjects, in this paper we also describe the safety concerns associated with transmitting and storing potentially sensitive personal data.

## 1. Introduction

The outbreak of the COVID-19 pandemic presented extraordinary challenges for the healthcare system [1]. Shielding patients, especially the elderly, from potential sources of infection while providing them the cardiological care they need has become extremely difficult. Cardiology is one of the areas of medicine where electric and electronic devices were first used in both diagnostics and therapy, and where basic and relatively easy-to-acquire physical signals give useful, or even crucial, information about a patient’s condition (e.g., arterial blood pressure, heart rhythm). As soon as transmitting such information to a remote location became possible, the concepts of telemonitoring and even treatment at distance began to develop [2]. In the following article we aim to present a general overview of what has been achieved so far, as well as the persisting issues in the growing field of telemedicine in cardiology.

## 2. Aim and Methods

Aim: The aim of this narrative review is to summarize the current knowledge of the rapidly evolving field of remote monitoring in cardiology.

Material and methods: A literature search for original and review articles, advisories from professionals and position papers published since 2010 was performed using PubMed and Google search engines. Query terms “remote monitoring” and “telemedicine” were used in combination with “cardiology”, “heart failure”, and “arrythmia”.

## 3. Cardiac Implantable Electronic Devices

Permanent implantable electronic devices have been used in cardiology for many decades—at first cardiac pacemakers, then implanted cardiac defibrillators (ICDs) and cardiac resynchronization therapy devices (CRT-Ds). The most straightforward approach to remote monitoring (RM) is to acquire data from an already implanted device via a transmitter (provided for the patient) and communication technology that allows remote data transfer. Recently, a smartphone functionality has been developed that enables communication with implanted devices via Bluetooth [3]. For many years such data acquisition has been used to detect technical problems such as low battery level, electrode dysfunction, or insulation defects. At present, the device may also be used to assess patients’ clinical status, such as changes in heart rate (including the onset of arrythmias), respiration rate, or physical activity. RM may lead to the further optimization of implantable cardioverter-defibrillator leads and lower chances of inappropriate shock [4].

In theory, remote monitoring strategies might result in fewer hospital or clinic visits (only when necessary vs. according to schedule) and in the timely detection of adverse events [5]. Consequently, RM should result in a lower cost of care and in higher survival rates or, at least, in a better quality of life. Most experts agree that such transmissions should take place every 3 months, which is more often than recommended follow-up visits [6]. The type of data transmitted and the frequency of transmission depend on the type of device, the indications for its implantation (secondary vs. primary prevention), and patient’s clinical status.

Results from several clinical studies seem to support this concept. The IN-TIME trial is one of the few that demonstrated actual clinical benefits to remotely monitored patients (lower mortality). The patients who benefited the most were those with a history of atrial fibrillation; indeed, an onset of atrial fibrillation was the event that most frequently led to medical intervention [7]. In a meta-analysis of patients with heart failure (HF) and an ICD with telemonitoring function, all-cause mortality and hospitalizations were significantly reduced [8]. Similar results were provided by ALTITUDE and EFFECT studies [9,10]. In accordance with the above findings, the remote monitoring of implantable devices is indicated in cases of suspected AF, in patients with heart failure and low ejection fraction, and when there are known technical problems with the device or any of its components.

Remote monitoring was also found to result in lower costs without compromising patient survival [11]. This finding in itself is a very important one given the ever-increasing number of HF patients and the amount of human and economic resources necessary to diagnose and treat them. Any solution that might lead to lower costs or better resources allocation would result in enormous savings on a regional or country level [12]. The standardization of the data recorded by devices from different manufacturers is a persisting issue. It is necessary to allow uncomplicated access to the data for all treating physicians as the inability of some health professionals to acquire certain data may compromise patient safety [13].

## 4. Wearable Devices

Another approach to RM is to use an external wearable device that has been specifically designed for this purpose. Recent advances in chipset electronics and sensor technology made such devices both affordable and efficient. They can provide information that leads to medical intervention or hospitalization. Additionally, they make remote care possible, thus increasing healthcare standards for rural populations [14]. A system of remote monitoring via a wearable device is usually viewed as consisting of four elements: the device itself, a network, a communications interface that allows data transfer, and an analytics platform that integrates large amounts of data and identifies crucial information [15]. The first wearable devices used for cardiac RM were Holter monitors. At present, their size and limited time of operation makes them much less efficient than more recent designs. The new solutions can be further divided into two groups—those that combine a sensor and a remote signal transmitter, and those that require another device (usually a smartphone) for remote data transmission.

The design of sensors used for wearable devices may vary, but they usually consist of patches or wrist bands, and sometimes of a phone-connected probe. Most often they register electrocardiogram (ECG) and blood pressure, and in some instances also oxygen saturation. Apart from standard ECG signal, the most commonly used method to register heart rate is the ballistocardiogram, which detects the repetitive movement of the body caused by blood ejection with each heartbeat. Other techniques such as phonocardiography and seismocardiography are also used, and may provide meaningful signals [16]. There is also a system for registering ECG via a single chest lead. It has proved to be more efficient in detecting arrythmias than standard Holter monitoring (possibly due to its 14-day battery life). This system only allows data to be analyzed after the recording period has been finished [17].

There are also several commercially available devices that enable data to be analyzed in real time for both the remote physician and for the patient. In the Apple Heart Study (performed from 2017 to 2019), a smartwatch-based system successfully identified irregular heart rates in 0.5% of the study population (consisting of 419,237 patients). The subsequent patch-based remote ECG monitoring (successfully performed in 450 patients with irregular HR) revealed AF in 33% of those cases [18,19]. The progress made in the field of wearable devices may be illustrated by the introduction of ECG monitoring textiles [20] and then wearable cardioverter-defibrillators that may be better suited to some subsets of patients than conventional ICDs [21].

## 5. Smartphone-Based Systems

ECG, blood pressure (BP), heart rate (HR), and body weight data may be obtained by an external device and transferred to a compatible smartphone. Additionally, smartphone manufacturers also offer HR and BP measurements using only the phone camera. This method uses photoplethysmography (infrared light absorbed differently by different tissues). The detector is able to measure the amount of blood flowing through the arteries. The HR measurements thus obtained have proved reliable and correlated well with ECG [22].

There are smartphone-based designs (using diode-camera sensor systems) that have demonstrated over 90% accuracy, sensitivity, and specificity in detecting atrial fibrillation, as well as premature atrial and premature ventricular contractions [23,24]. Smartphone-camera-based and probe-based smartphone pulse oximetry systems also proved to be non-inferior to standard methods [25]. Although the classic inflatable cuff remains a gold standard for BP measurement, several smartphone-based designs exist that use various types of sensors with astonishingly accurate results. They include accelerometers, finger sensors, microfluidic sensors, as well as seismo- and ballistocardiography [26,27]. Attempts have also been made to use data derived from RM systems to estimate cardiorespiratory fitness and stress exposure—important parameters in quantifying the risk of heart failure and ischemic events, respectively [28,29].

In a survey conducted by Sohn et al., 60% of HF patients expressed interest in a smartphone app designed for them. Interest in the device was correlated with more advanced stage of the disease and negatively correlated with patient age [30]. In a study conducted among healthcare providers in Australia, key elements of a smartphone-based system for monitoring patients after acute coronary syndrome (ACS) were identified. They included education on diet and symptoms, the measurement of weight and blood pressure, as well as the monitoring of pain and emotional status. The participants identified the importance of real-time video conferencing. Old age and low education levels were identified as potential obstacles to the widespread use of smartphone-based systems [31].

The highest levels of diagnostic accuracy, as in some of the above-presented examples, have been obtained using both high-end modern (and therefore more expensive) smartphones and very sophisticated data-processing algorithms (including machine learning, etc., to discern signal from artifact) [32]. It should be expected that in real-life conditions (e.g., the widespread use of older, simpler, and less-expensive smartphone designs) the data quality would be lower. Consequently, blood pressure measurement using only a smartphone is still judged unreliable and prone to errors in the scientific literature [33]. As of 2016, 99% of smartphone applications were not considered medical devices, and consequently not regulated by the FDA.

## 6. Benefits of Remote Monitoring—Data from Clinical Trials

As in every other field of medicine, evidence from clinical trials is necessary to reliably assess treatments given or diagnostic modalities in terms of benefits for the patient—if not in terms of survival than at least regarding quality of life. In the sections below we present data concerning three main areas of cardiology where the evidence in support of RM seems to be the strongest: cardiovascular risk factor management, detection and treatment of arrythmias, and monitoring heart-failure patients.

### 6.1. Risk Factor Management

Given the enormous number of deaths that can be attributed to coronary artery disease, the identification and (when possible) modification of well-known predisposing factors of atherosclerosis has been proposed as the most effective prevention strategy. Several cardiovascular risk factors, such as blood pressure, body weight, or level of physical activity, are relatively easy to measure, quantify, and modify—all of which should make them ideal targets for remote monitoring and intervention.

Mobile health interventions were positively correlated with lower BP, smoking cessation, and increased physical activity, which are all considered crucial risk factors of coronary artery disease [34]. In some studies, teleconsultations led to measurable reductions of coronary artery disease risk factors [35]. In a study by Margolis et al. [36], remotely controlled blood pressure measurements resulted in lower blood pressure values compared to standard care.

A review by Burke et al. [37] presents trials with successful interventions via smartphone apps for weight reduction, lipid levels, blood pressure, physical activity, and smoking cessation. However, it should be noted that in many cases an additional device was required (e.g., ECG apparatus). Efficient communication by means of voice transmission, website, message transmission, or face-to face was crucial for the intervention to be efficient. Unfortunately, diverse methodologies and small sample sizes (many of the studies were pilot/feasibility studies) makes it difficult to compare or generalize their results. It should also be noted that addressing only one coronary artery disease risk factor (which is what most of the apps do) is never sufficient to substantially reduce an increased risk of the disease.

### 6.2. Arrythmia Detection and Management

ECG registration and analysis was one of the first methods tested in the setting of remote management. As the automatic identification of cardiac rhythm can now be performed with a high degree of confidence, it is expected that arrythmia detection might be successfully carried out via RM.

In the REM HF trial (carried out in 2011–2016 on patients with HF and an implanted device), the remote monitoring of patients with atrial fibrillation (AF) was compared to standard care [38]. The use of RM resulted in more interventions (visits, hospitalizations, etc.), but no differences in mortality between the RM group and standard care group were detected. In the sub-analysis of the AF patients from this trial, several parameters were taken into consideration. The AF detected was qualified as none, paroxysmal, or continuous. The AF burden (amount of time spent with AF) and the incidence of subclinical AF (not felt by the patient) were also analyzed [39].

However, the use anticoagulation and the incidence of thromboembolic events were not specifically analyzed. As pointed out by experts who analyzed the results of REM HF trial, it is reasonable to think that AF patients do benefit from RM, but the benefits are not the type of events that were prespecified to be analyzed in this trial [40]. Such potential effects might include, for example, the earlier detection of new or subclinical AF, better assessment of symptoms potentially related to AF, more detailed assessments of potential heart failure worsening, and more detailed analysis of the ratio of biventricular pacing in patients with CRT-D [41].

It has been demonstrated that RM is superior to conventional Holter monitoring in diagnosing AF in patients after stroke or TIA [42]. However, not all atrial high-rate episodes detected by a single lead represent AF. Whether patients with these arrythmias detected only by RM are indeed at higher risk of stroke and should be given anticoagulant treatment has not yet been determined [43]. There are ongoing trials designed to answer this question [44], but so far the significance of silent AF is still unknown [45]. According to experts, remote ECG monitoring is particularly beneficial for patients with rarely occurring symptoms (less than once a day) that may be caused by arrythmias [46].

### 6.3. Monitoring Heart Failure Patients

There have been a number of studies evaluating RM of HF patients. In many of them, unique devices or analytical tools that were designed for this purpose were used. As an example, a special Heart Logic algorithm was developed to diagnose conditions related to HF. The system proved able to detect clinically relevant events, and the alert-based strategy seemed more efficient than a standard schedule-based follow-up scheme [47]. Similarly, in the Triage HF plus program from 2018, a wearable device combined with a telephone interview proved to be a feasible and useful strategy for HF patients [48].

TIM HF and TIM HF2 were both large trials evaluating the remote monitoring of heart failure patients. In TIM HF, completed in 2011, the remote monitoring of ambulatory patients with chronic HF was not associated with a reduction in all-cause mortality when compared to standard care [49]. The TIM HF2 trial (performed between 2013 and 2017) suggests that a structured remote patient management intervention, when used in a well-defined heart failure population, could reduce the percentage of days lost due to unplanned cardiovascular hospital admissions and all-cause mortality [50].

In the largest trial evaluating the benefit of RM in HF patients (REM HF), no significant differences were found between patients using RM and those in the standard care plan. The endpoints analyzed in this study included death and hospitalization (resulting from both cardiovascular and non-cardiovascular causes) [38]. Remote monitoring was also compared to standard care in the RESULT study. Both groups demonstrated similar mortality but the incidence of hospitalizations was lower in the remote care group [51].

A program of hybrid rehabilitation consisting of standard visits and remote monitoring of training was tested in a study by Piotrowicz et al. in 2019 [52]. It proved more effective than the standard care (in terms of increased oxygen consumption and the score in a quality-of-life questionnaire). The mortality and hospitalization rates were not different from a group that underwent a standard rehabilitation program. Hybrid telerehabilitation is safe and feasible for patients with various levels of HF [53]. It is the only safe solution (and therefore recommended) during the COVID-19 pandemic [54]. On the other hand, in the BEAT HF study of patients hospitalized for HF (carried out in 2013–2014), combined health coaching telephone calls and telemonitoring did not reduce 180-day readmissions [55].

In some cases of symptomatic HF patients, the invasive monitoring of hemodynamic parameters is indicated. The CardioMEMS heart failure sensor (Abbott Vascular) is an implanted device that measures pulmonary artery pressure and sends the results to an external transmitter. Its use proved to be safe, reduced the number of hospitalizations, and resulted in a better quality of life [56,57]. Other implantable devices measuring right ventricular pressure and left atrial pressure have also been proposed [58]. Another approach that is currently being evaluated is the noninvasive measurement of hemodynamic parameters as a function of thoracic impedance [59].

From the studies listed above it may be concluded that there are no clear benefits of RM in terms of overall mortality (with the notable exception of the TIM HF2 study). Several reasons have been postulated, including patient selection (inclusion of low-risk patients with low event rate) and compliance issues. Nevertheless, for some subgroups or perhaps even most patients, RM may offer the advantage of improved life quality, for example due to less-frequent hospitalization or clinic visits. It seems that well-structured remote monitoring programs, such as hybrid designs consisting of standard visits and remote care, might be the most useful.

### 6.4. Monitoring for Ischemia and Acute Coronary Syndromes

Ischemia and acute coronary syndromes are, at least in some cases, relatively easy to detect in automatic ECG analysis. It is therefore worth noting that so far there is no proof that remote monitoring for ischemic changes is beneficial for patients. For instance, in a study conducted by Saleem et al., in-patient telemetry rendered no useful information for predicting short-term coronary events or mortality, or for predicting long-term mortality in low-risk patients hospitalized with chest pain [60]. It is conceivable that the detection of a single acute coronary event may require the monitoring of a large group of patients for a long time, which lies in contrast to the detection of AF episodes (which in some patients may occur several times a week). Therefore, it may be expected that studies of large groups surveyed for a long time might indeed prove the validity of the concept.

## 7. Devices and Programs for Prespecified Groups of Patients

It is worth noting that some manufacturers, in collaboration with clinical centers, have developed devices or sets of devices specially designed for the remote monitoring of prespecified subgroups of cardiological patients. A system of several devices including blood pressure monitor, thermometer, weight scale, step count watch, single-lead ECG device, 12-lead ECG device, and pulse oximeter has been proposed to remotely monitor post coronary bypass graft patients. The trial is primarily designed to detect post-operative AF, and preliminary results are expected this year [61]. Similarly, a smartphone-based program was proposed for post ACS patients. It includes pain and body weight monitoring, dietary consultations, and online communication with a healthcare professional [31]. Such devices used in selected groups have the potential to demonstrate actual benefits for the groups in which they are used. Even if we still lack convincing data, it is to be expected that in this setting—a device and/or a program developed for monitoring in a specific clinical situation—the real advantages of RM might be documented first.

## 8. Practical Benefits

Apart from strictly medical benefits (or lack thereof), another issue to consider is the impact of RM monitoring on the practical aspects of healthcare. There are many examples of remote care schemes that had major influences on the everyday functioning of healthcare systems. A novel remote monitoring scheme resulted in more monitoring-related visits and fewer unscheduled visits. This enabled better human resources allocation and increased standard of care in the center that conducted the study [62]. E-consults offer the advantages of being less expensive and consuming less time and human resources than standard consultations. On the other hand, the quality of these procedures needs to be regularly monitored, and they provide less opportunity for staff training [63].

In a survey conducted in Italy, telemedicine was used in 84% of pre-hospital ECG analyses. Remote control of ICD/CRT–D was performed in 42% of cases, and HF patient monitoring in 37% of cases [64]. In another Italian survey there was a marked increase in the use of telemonitoring between 2012 and 2017 (in terms of number of patients per center) [65]. A hospital–community–family-based telehealth study was determined to be feasible in a study conducted by a Chinese group. The program was perceived as effective and satisfactory by both patients and physicians who participated [66].

Detailed data covering the use and readiness to use RM systems vary between countries. As an example, as recent as 2015, two-thirds of the US population owned a smartphone [67]. Even higher numbers are reported for several developed countries in more recent surveys [68]. On the other hand, only half of the people who have a wearable cardiac monitoring device reported using it on a daily basis [69]. Among the elderly, the percentage is probably even lower. Nevertheless, 60% of them would like to use this type of device in the future [70]. Many studies demonstrate that patients are indeed willing to use wearable RM devices if they are lightweight and easy to operate, especially when given a clear benefit such as lower insurance costs [71]. A study conducted at Mayo Clinic demonstrated that only 20% of patients participating in a smartphone-based study for recording BP and weight required readmission within 3 months, as opposed to 60% of those who did not participate in the program [72].

As a consequence of the rapid development of RM technologies, position papers were issued by the American Heart Association (AHA) [73] and the European Society of Cardiology (ESC) [74]. In Europe, several country-level cardiological societies published their position papers, some including recommendations with levels of evidence (I, IIa, IIb). Examples include Germany [75], Austria [76], and Poland [3]. Possible limitations of remote monitoring are a lack of standardization and the potential vulnerability of the stored data, especially because smartphone apps are in most cases not regulated by any authority. There is a risk of the data being sold to third parties by app developers. There are several important factors limiting the development of remote care. First, the quality of technology is often insufficient for reliable signal measurement and transmission. Second, most of the current legislation is based on traditional medicine and therefore inadequate for remote technologies. Finally, despite the emerging evidence of RM’s efficacy in large cohort studies [77,78], there is still no reimbursement of such methods [79].

## 9. Safety Concerns

As with every new medical technology, there are concerns regarding the safety of remote monitoring. Even if most of the apps are not medical devices, this does not mean that they do not collect medical data. In a study conducted by Sunyaev et al., it was found that only 30% of the 600 most popular health apps had a privacy policy. Even more alarmingly, nine out of ten of the most downloaded health and fitness apps were found to sell data to third-party domains. Although some of the data transferred may be used for benign reasons (for example allowing local communities to decide where to build jogging or bike tracks), unfortunately they may also be used by insurance companies to differentiate insurance premiums between customers [80].

Concerns about data security might be one of the most important factors limiting more widespread use of these apps [81]. The security of remote access to ICDs is an issue that cannot be overlooked. Some functions of the device (e.g., device-induced tachycardia) can potentially be hazardous to patients. The feasibility of radio-based attacks on an implanted device (including commanding a shock) was proven by a team of researchers [82].

## 10. Conclusions

Remote monitoring is a novel, modern, and very promising addition to standard cardiological care. Despite enormous technological progress in signal acquisition, data transfer, and analysis, there are several issues that remain a challenge. One of these challenges is the standardization of both the devices and the data. Identifying the groups of patients that would benefit the most from remote care is another problem. In the trials conducted so far there is indeed growing evidence of the benefits of remote monitoring in several clinical settings. Nevertheless, the scientific data collected so far for other subgroups are insufficient to draw clear conclusions. Safety issues should not be overlooked given the large amounts of potentially sensitive data that may be processed. Two facts seem to be clear though: there is an enormous interest both from patients and healthcare professionals, and there are huge potential benefits in terms of cost effectiveness and quality of life.

## Data Availability

Not applicable.

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
