# Peer review of "Telemedicine in Cardiology: Modern Technologies to Improve Cardiovascular Patients’ Outcomes—A Narrative Review"

_medicina, 2022, doi:10.3390/medicina58020210_

Round 1

Reviewer 1 Report

General: 

This review presents an overview of the status quo of monitoring-technologies for patients with cardiovascular diseases and risks for developing it. However, the manuscript is missing transparency in terms of the specific aim of the review as well as the methods the authors used. It is unclear according to which criteria the publications are examined. Therefore, a reproduction of the outcome is currently not possible.

Additionally an adoption of IMRAD structure (Introduction, Method, Results, Discussion, Conclusion) would improve both the readability and traceability for reproduction. Due to the lack of separation between results and discussion, it is not clear whether these are opinions and evaluations of the authors or statements from referenced papers. It is not recognizable which methodology the structure of the "results" follows. 

As the aim of this review is not clearly and precisely stated, it stays unclear which current gap of research this review fills. It is also unclear how this reviews is distinguished from other reviews or systematic reviews and what sets it apart. It also remains unclear whether the own scientific question could be answered within the scope of the review. 

Specific: 

  • Abstract: Misses aim of the review, the method it follows as well as the main findings
  • Introduction: Goal of the review not clearly defined. Please outline more specificly on the aims of the review.
  • Results: Results not recognizable as such. Structure of the results section does not contain a recognizable thread (Example: Paragraph 5. Benefits of Remote monitoring [...] comes before the use-case of monitoring 8. Monitoring of heart failure patients). I strongly recommend to restructure the manuscript according to IMRAD.
  • line 48-49: Missing reference
  • 61-63: Contradicts statement from conclusion: "Identifying the groups of patients that would benefit the 342 most from remote care is another problem, especially in view of somewhat unclear evidence emerging from trials conducted so far. "
  • 67-71: Reference [10] is most likely not the primary source for the statement it should prove. Further reference tracking recommended. 
  • 78-81: Very imprecise: Which "advances" and which "information"?
  • 129: Wrong spelling: et al.
  • 141: Wrong spelling: etc. 
  • 144: Missing Comma: "As of 2016, 99%.."
  • 158: Easy according to whom/ in relation to what? Define "easy" or add a reference.
  • 172-175: Statement belongs in a discussion since it involves an interpretation of the authors. If it refers to a statement from a reference, it should be marked as such. 
  • 181-182: Missing reference for REM HF trial. 
  • 177-195: Insufficient reference density. 
  • 256: Wrong spelling: et al. 
  • 263: Dot at the end of the heading.
  • 277: Paragraph should be part of a discussion and marked as such.
  • 317-319: Evidence from large cohorts exists and can be found in a systematic review we conducted (Kinast, B.; Lutz, M.; Schreiweis, B. (2021): https://doi.org/10.3390/)
  • 320: Dot at the end of the heading.
  • 338: Dot at the end of the heading.
  • 331: Spell check: [...] might BE one“
  • 337: Interesting fun fact but does not add anything to the body of knowledge. I'd rather remove it from the manuscript.
  • 342-344: Controversial statement that may apply to the publications examined in the review, but cannot be generalized because it is not a systematic search. It is correct that most of the existing studies are not comparable and can only be considered in isolation, yet this is evidence. Furthermore, there is a lack of meta-analyses, e.g., on telemonitoring of cardiovascular diseases via wearables due to this. 

Author Response

please see the attachement

Reviewer 2 Report

This review manuscript focuses on application of the modern technologies for telemedicine. The manuscript describes advantages and disadvantages of modern technologies and compares them with “old” techniques.

Comments:

  1. I found that list of abbreviations is not complete (TIM, REM,...)
  2. Citations Margolis et al. and Burke et al. is not right. It should be Margolis et al. [33], Burke et al. [34]
  3. After minor revisions I recommend it for publication.

Author Response

please see the attachement

Reviewer 3 Report

The work is good, and the scope of journal and the manuscript has a match. I feel the work done is good, need to include the following suggestions for betterment if suitable.

  1. The Manuscript is theory oriented. Please consider to add a block diagram or a flow diagram to be better readable
  2. Section I is very limited, add more content related to cardio issues such as stroke and its statistics with reference to WHO reports, since the focus is drawn parallel to COVID-19, Please consider depression, stress and bad diet under the causes (refer the work: "Neural Network Based Mental Depression Identification and Sentiments Classification Technique From Speech Signals: A COVID-19 Focused Pandemic Study." Frontiers in Public Health 9 (2021). )
  3. From Section 2 and Section 8, the minimum size of a paragraph should be 6 to 8 lines. please align it accordingly
  4. Avoid direct reference of products / company in section 3 e.g. Apple Watch. If mentioning is inevitable, please provide the reference for the same
  5. Consider to cite, recent development as [13][14] 
  6. Consider the following works in ECG related validations
    1. Dagher, Lilas, Hanyuan Shi, Yan Zhao, and Nassir F. Marrouche. "Wearables in cardiology: Here to stay." Heart Rhythm 17, no. 5 (2020): 889-895.
    2. Ahmed, S. Syed Thouheed, K. Thanuja, Nirmala S. Guptha, and Sai Narasimha. "Telemedicine approach for remote patient monitoring system using smart phones with an economical hardware kit." In 2016 international conference on computing technologies and intelligent data engineering (ICCTIDE'16), pp. 1-4. IEEE, 2016.
    3. Hong, Shenda, Yuxi Zhou, Junyuan Shang, Cao Xiao, and Jimeng Sun. "Opportunities and challenges of deep learning methods for electrocardiogram data: A systematic review." Computers in Biology and Medicine 122 (2020): 103801.
  7. The conclusion drawn is unclear. Please use a table representation for clear drafting of research reviews.

Author Response

please see the attachemnet

Round 2

Reviewer 1 Report

General:

Thank you very much for your detailed reply. I would also like to thank you for taking some of my suggestions into account, which I believe will benefit your publication in terms of readability. The deeper subdivision as it has now been implemented makes a significant contribution to this. 

Since the impetus for writing this narrative review was to show the current status, it would be useful if the authors point out the investigation period (as part of the method) and state the year in which each included study was conducted so that the reader could better understand the timeliness of these findings. 

As of for now, it stays intransparent which objectives, inclusion criteria or general method this research follows and if the research guiding question could be answered. My previous request for the development of a methodology section was unfortunately not followed up. 

I would encourage the authors to add a paragraph to the publication outlining the goals and carefully describing the method - even if it is a narrative review, which is currently not clear from the text - to achieve a more transparent research structure.

As a final suggestion, I would recommend to add in the title that this is a narrative review so that it is clearly distinguished from other work. 

Specific:

131-132: Add year when the study was performed.

132-134: Consistent use of ratios. (0,5%; 1/3). 

131-134: For a comparison please add size of the study populations. 

190: Benefits of Remote Monitoring - […]

279-280: Remove brackets

354-356: When giving the US population as an example it would help the readers' understanding if another country was shown for comparison. 
